# Comparative Study on Violent Sloshing with Water Jet Flows by Using the ISPH Method

**Hua Jiang [1,2], Yi You [1], Zhenhong Hu [1,*], Xing Zheng [1] and Qingwei Ma [1,3]**

1   College of Shipbuilding Engineering, Harbin Engineering University, Harbin 150001, China;
    jiangh@gumeco.com (H.J.); youyione001@hrbeu.edu.cn (Y.Y.); zhengxing@hrbeu.edu.cn (X.Z.);
    q.ma@city.ac.uk (Q.M.)
2   Guangzhou Marine Engineering Corporation, Guangzhou 510250, China
3   School of Mathematics, Computer Science & Engineering, City, University of London, London EC1V 0HB,
    UK
*   Correspondence: huzhenhong@hrbeu.edu.cn; Tel.: +86-451-8256-8147

**Abstract:** The smoothed particle hydrodynamics (SPH) method has been playing a more and more important role in violent flow simulations since it is easy to deal with the large deformation and breaking flows from its Lagrangian particle characteristics. In this paper, the incompressible SPH (ISPH) method was used to simulate the liquid sloshing in a 2D tank with water jet flows. The study compares the liquid sloshing under different water jet conditions to analyze the effects of the excitation frequency and the water jet on impact pressure. The results demonstrate that the water jet flows can significantly affect the impact pressures on the wall caused by violent sloshing. The main purpose of the paper is to test the ISPH ability for this study and some useful regulars that are obtained from different numerical cases and study the effect of their practical importance.

**Keywords:** ISPH; liquid sloshing; water jet flow; impact pressure; excitation frequency

## 1. Introduction

The phenomenon of violent sloshing appears widely in the field of Naval Architecture and Ocean Engineering, especially in liquid cargo carriers, such as the LNG (liquefied natural gas) carriers. During this process, the motion of liquid in partially filled tanks may cause large global and local loads on the tank walls when the frequency of sloshing is close to the natural frequency of the liquid tank. This consequence would be very serious in engineering practice, which may cause damage to the hull structure and even affect the stability of the carrier [1]. Therefore, sloshing is an old topic, but it still needs to be studied in depth.

At present, lots of studies on the violent sloshing flows have been carried out by the linear and nonlinear potential flow theory and the scaled model experiment. Faltinsen [2–4] used the incompressible potential flow theory to simulate liquid sloshing and obtained the formulas that have been widely used in the field of sloshing simulation. However, the method can be used to study sloshing tanks with relatively simple geometry and internal structure. In addition, Akyildiz et al. [5] investigated the pressure distribution on a rectangular tank during the process of sloshing by an experimental method. Sames et al. [6] studied sloshing in a rectangular tank with a baffle, and a cylindrical tank was also considered. Indeed, the experimental method can be applied to study the sloshing in the tank with more complex shapes, but it also requires high expenses for the site and facility. Hence, the numerical method has been getting more important in the simulation of liquid sloshing in recent years. The conventional numerical methods are carried out by using Euler grids. Wu et al. [7] simulated the sloshing waves in a 3D tank based on the finite element method (FEM).

In the conventional grid-based methods, in order to track the moving free surface, some additional techniques, such as the Volume-of-Fluids, are used in the methods. The VOF uses the volume fraction of fluid in gird to define the free surface. However, the problems of numerical diffusion become serious when the surface cell becomes extremely complicated, such as in a liquid sloshing, which can easily fail to simulate because of the large deformation of grids. Recently, a kind of mesh-less method named smoothed particle hydrodynamics (SPH) has attracted quite a few researchers' attention [8]. It does not depend on any grids, and the computation is purely based on a group of discrete points that can move freely. So, it can capture the free surface flow conveniently, which is more suitable for treating the problems of large deformation of free-surfaces. Delorme and Colagrossi et al. [9] investigated impact pressure in the case of shallow water sloshing by the SPH method, compared the results with experimental ones, and then discussed the influence of viscosity and density re-initialization on the SPH results. Gotoh and Khayyer [10] simulated the violent sloshing flows using the incompressible SPH (ISPH) method and presented two schemes to enhance the accuracy of the simulation of impact pressures. Zheng and You [11] compared the effect of different baffle configurations on mitigating sloshing by the ISPH method. A great deal of research [12,13] shows that the ISPH method can improve the accuracy and stability of the calculation pressure, and the pressure field is smoother.

As a matter of fact, the marine environment is always very complex when sloshing happens. If an oil fire occurs, it would be a big disaster and hard to control, and the water jet flow outside would get into the tank to put out the oil tank fire, which would influence the impact effect of sloshing. Such a consequence may cause more serious damage to the hull structure, which would be very serious in engineering practice. With regard as the problem of jet flows, Hatton et al. [14,15] studied the trajectories of large water jets that are used in the design of fire-fighting systems, particularly those used in offshore situations, and evaluated the effects of flow-rate, pressure, and nozzle size during the process of the system design. Fischer et al. [16] used three different CFD codes, namely, the CHYMES multiphase flow model, the FEAT finite element code, and the Harwell-FLOw3D finite volume code, to simulate the problem of a laminar jet of fluid injected into a tank of fluid at rest and make a detailed comparison. Aristodemo et al. [17] studied the plane jets propagating into still fluid tanks and current flows by using the WCSPH method. Andreopoulos et al. [18] carried out an experiment on the flow generated by a plane with a buoyant jet discharging vertically into shallow water.

In this paper, the liquid sloshing with a water jet flow from the top of the tank will be studied by using the incompressible SPH (ISPH) method. Through the comparison of different situations, the sloshing effects and characteristics of the impact pressure are studied. The aim of this study is to summarize the influence of the water jet flow on sloshing, so as to give a reference for practical engineering.

## 2. ISPH Methodology

### 2.1. Governing Equations

The SPH model is based on the semi-Lagrangian form of the continuity equation and the momentum equation. In the ISPH method, the density of fluid is considered to be a constant, and thus, the governing equations are written as follows:

$$\nabla \cdot \boldsymbol{u} = 0, \tag{1}$$

$$\frac{D\boldsymbol{u}}{Dt} = -\frac{1}{\rho}\nabla P + \boldsymbol{g} + \nu_0\nabla^2\boldsymbol{u}, \tag{2}$$

where $\rho$ is the density of fluid; $\boldsymbol{u}$ is the velocity of particle; $t$ is the time; $P$ is the particle pressure; $\boldsymbol{g}$ is the gravitational acceleration; $\nu_0$ is the kinematic viscosity; and $\nabla$ is Hamilton operator, which is a vector operator.

## 2.2. Particle Approximation

The computational domain of the SPH method is composed of a group of discrete particles, and each particle is given corresponding physical information, such as density, volume, mass, velocity, and pressure. The physical information of each particle can be approximately obtained by the information carried by the surrounding particles, which is shown as follows.

$$f(\boldsymbol{r_i}) \ = \ \sum_{j=1}^{N} \frac{m_j}{\rho_j} f\left(\boldsymbol{r_j}\right) W\left(\boldsymbol{r_{ij}}\right) \tag{3}$$

where *f(r)* represents the physical information of particles, *m* is the mass of the particle, *i* and *j* are the center particle and neighbor particle, respectively. *N* is the number of neighbor particles. *W(r$_{ij}$)* is the kernel function, which can reflect the different effects between different particles. In this paper, the cubic B-spline kernel proposed by Monaghan et al. [19] is used as follows:

$$W\left(r_{ij},\,h\right) \ = \ \alpha_d \begin{cases} \frac{2}{3} - q^2 + \frac{1}{2}q^3, \ 0 \le q < 1 \\ \frac{1}{6}(2-q)^3, \ 1 \le q < 2 \\ 0, \ 2 \le q \end{cases} \tag{4}$$

where *h* is the kernel smoothing length, $r_{ij}$ is the distance between the *i* and *j* particle, $\alpha_d$ is a constant, and when the case is 2D, its value is $\frac{15}{7\pi h^2}$ and $q = \frac{r_{ij}}{h}$.

So the derivatives of *f(r)* can be represented as:

$$\nabla f(\boldsymbol{r_i}) \ = \ \sum_{j=1}^{N} \frac{m_j}{\rho_j} f\left(\boldsymbol{r_j}\right) \nabla_i W\left(\boldsymbol{r_{ij}}\right), \tag{5}$$

where $\nabla_i$ is the gradient, which is taken with respect to the particle *i*.

## 2.3. Poisson Pressure Equation

In the ISPH, a two-step projection method is used to solve the velocity and pressure field from the continuity equation and momentum equation [20]. The first step is the prediction of velocity without considering the pressure term. The second step is the correction step in which the pressure term is added through the pressure Poisson equation (PPE), then the PPE is obtained as follows:

$$\nabla^2 P^{t+\Delta t} \ = \ \frac{\rho \nabla \cdot \boldsymbol{u^*}}{\Delta t} \tag{6}$$

where $\boldsymbol{u^*}$ is the intermediate particle velocity at the first step.

Similarly, Shao and Lo [20] proposed a projection-based incompressible approach by imposing the density invariance on each particle, leading to the following PPE equation:

$$\nabla \cdot \left(\frac{1}{\rho^*} \nabla P^{t+\Delta t}\right) \ = \ \frac{\rho_0 - \rho^*}{\rho_0 \Delta t^2} \tag{7}$$

where $\rho^*$ is the density at the intermediate time step, $\rho_0$ is the initial fluid density, and the combined PPE incorporates both the velocity-divergence-free condition and the zero-density-variation condition, which is obtained as:

$$\nabla^2 P^{t+\Delta t} \ = \ \alpha \frac{\rho_0 - \rho^*}{\Delta t^2} + (1-\alpha)\frac{\rho_0 \nabla \cdot \boldsymbol{u^*}}{\Delta t} \tag{8}$$

where $\alpha$ is a blending coefficient. If $\alpha$ is equal to 1, in Equation (8), the source term of PPE adopts the density variable effect, which may lead to substantial pressure noises and particle randomness caused by larger density changes. If $\alpha$ is equal to 0.0, the source term of PPE adopts the velocity divergence effects, which is smoother for source term distribution, but it will cause the pattern distribution of some

particles. In order to make the source term more reliable, $\alpha$ is equal to 0.01 in this paper, according to lots of computational experience. And more advanced PPEs with the error-compensating source term (ECS) can follow the references [21,22].

*2.4. Calculation of Spatial Derivatives*

According to Equation (5), the spatial derivatives of pressure and velocity can be calculated as follows:

$$\nabla P_i = \rho_i \sum_{j=1}^{N} m_j (\frac{P_j}{\rho_j^2} + \frac{P_i}{\rho_i^2}) \nabla_i W (\boldsymbol{r}_{ij}, h), \tag{9}$$

$$\nabla \boldsymbol{u}_i = -\frac{1}{\rho_i} \sum_{j=1}^{N} m_j (\boldsymbol{u}_i - \boldsymbol{u}_j) \nabla_i W (\boldsymbol{r}_{ij}, h). \tag{10}$$

Therefore, in the ISPH method, the viscous term adopts the following form:

$$\nabla (\nu_i \nabla \cdot \boldsymbol{u}_i) = \sum_{j=1}^{N} 4 m_j (\frac{\nu_i + \nu_j}{\rho_j + \rho_i} + \frac{\boldsymbol{u}_{ij} \, \boldsymbol{r}_{ij}}{\boldsymbol{r}_{ij}^2 + \eta^2}) \nabla_i W (\boldsymbol{r}_{ij}, h) \tag{11}$$

where $\eta$ is a small parameter to avoid the singularity, and in this paper, the value is chosen to be 0.1 $h$. So, the PPE is discretized by combining the SPH gradient and divergence rules to obtain:

$$\nabla (\frac{1}{\rho^*} \nabla P) = \sum_{j=1}^{N} m_j (\frac{8}{(\rho_j + \rho_i)^2} \frac{P_{ij} \, \boldsymbol{r}_{ij}}{\boldsymbol{r}_{ij}^2 + \eta^2}) \nabla_i W (\boldsymbol{r}_{ij}, h). \tag{12}$$

The treatment of free surface and solid boundary conditions follows the study of Zheng et al. [23].

*2.5. Inlet Boundary Treatment*

With regards to the case of a general closed boundary, sufficient particles are given at the initial time, and no particles are added during the middle steps. However, the model of sloshing with a water jet flow needs to add particles constantly at the top boundary. Hence, the condition of the top boundary should be treated as an inlet boundary to add the particles of the water jet. However, if the particles are added directly, there will be only a row of particles at the beginning, which would lead to big errors in the particle approximation.

In this paper, three rows of virtual particles are arranged at the top boundary, as shown in Figure 1a, which have the same velocity and physical information as the initial water jet particles, but they do not participate in the solving of PPE, they are just used in the particle approximation. The velocities of virtual particles remain constant, and when they have moved a distance of a particle size *dx,* as shown in Figure 1b, they will return to the initial position automatically, as shown in Figure 1c. So, the distance between virtual particles and the water particles is just enough to add a row of new water particles. Finally, the new row of water particles will be added, as shown in Figure 1d.

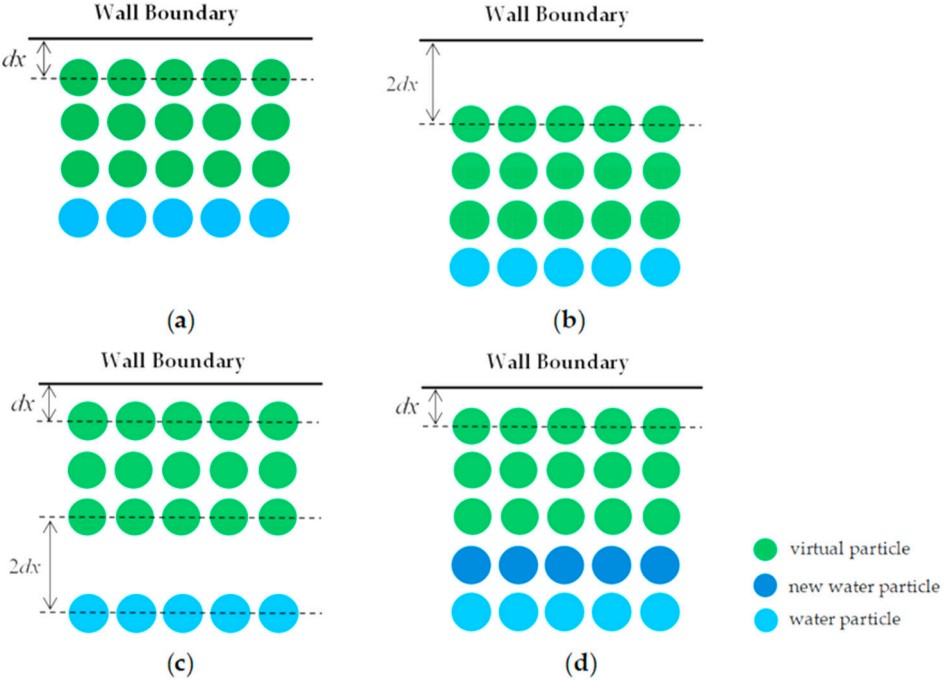

**Figure 1.** Different steps of the inlet boundary treatment: (**a**) Initial distribution of particles; (**b**) Particles has moved a distance of a particle size; (**c**) The virtual particles are return to the initial positions; (**d**) The new water particles are added.

## 3. Numerical Results and Validation

In this section, the numerical results of the sloshing tank and water jet calculated by the ISPH method are compared with the experimental data to validate the accuracy of the ISPH model.

### 3.1. Sloshing Tank Simulation and Validation

Figure 2 illustrates the rectangular sloshing tank that was used in the experiment of Liao and Hu [24], and the schematic view gives the geometrical dimensions of the sloshing tank; the length of the tank is 1.2 m, the height is 0.6 m, the initial depth of the liquid inside is 0.12 m, and the rotation center is located at the geometric center of the tank. $M_1$ and $M_2$ are two pressure sensors, which are used to record the pressure of the two points. In the experiment, the sloshing tank is imposed a rolling motion, and the amplitude and period of excitation are set as 10°and 1.85 s, respectively. The distribution of the particles in the ISPH computational model is uniform, with a particle spacing $dx$ = 0.002 m. A constant time step of $dt$ = 0.0003 s is used. The pressure data is extracted for 8 s after about twice the sloshing period $T$, which is relatively stable.

Figure 3 shows the snapshots of the sloshing motion of the ISPH model with the calculated pressure fields. Meanwhile, the corresponding experimental photographs are considered for the comparisons of the motion patterns. According to the comparisons, the ISPH model can get a good agreement on surface profile with the experimental results, and most of the features of the violent sloshing process have been captured by the ISPH model in a satisfactory manner. In addition, the ISPH method also provides a reliable regular distribution of pressure at the impact regions, and the pressure fields show a very stable pattern with little pressure noise.

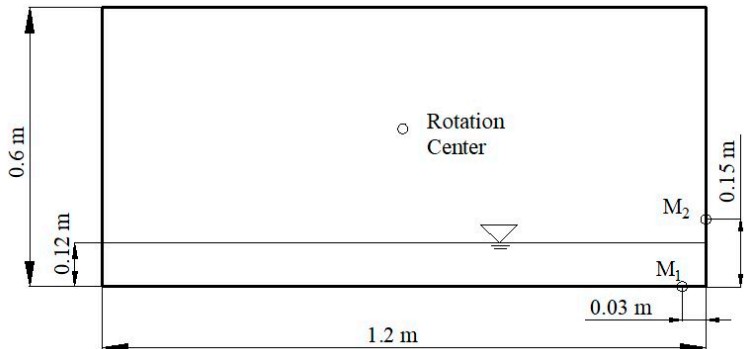

**Figure 2.** Schematic view of the sloshing tank following Liao and Hu [24].

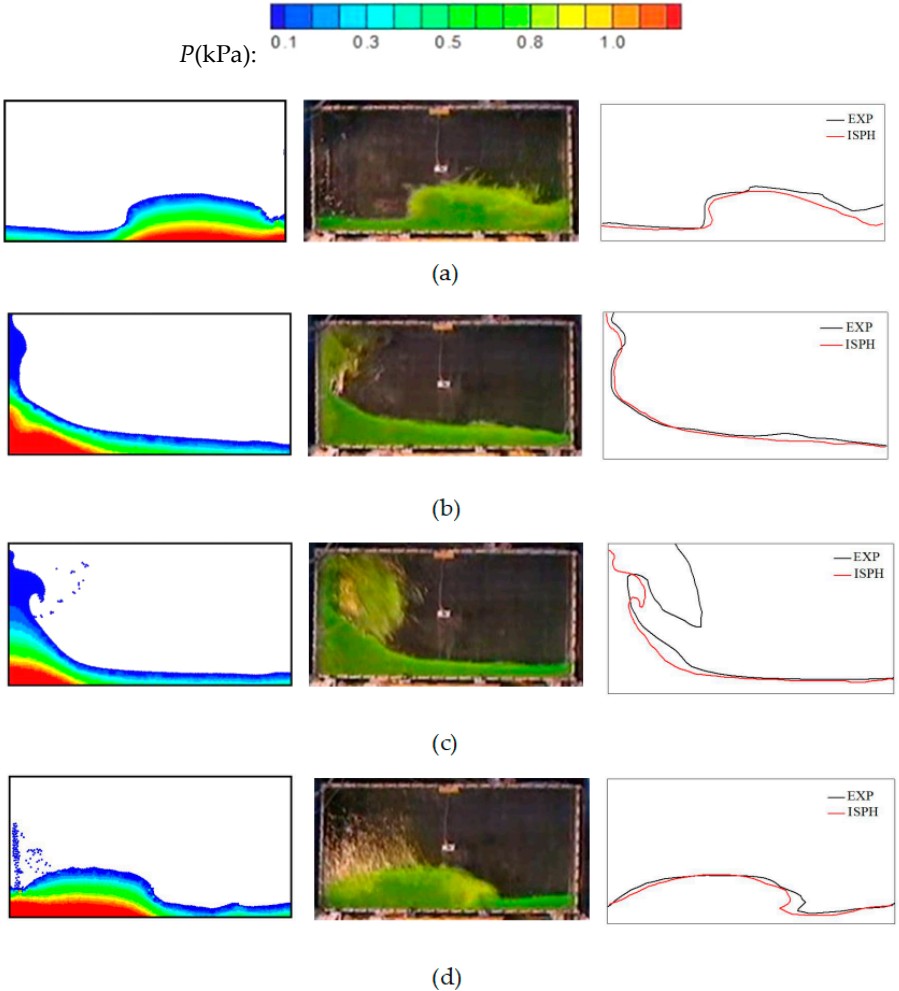

**Figure 3.** Sloshing snapshots computed by the ISPH (left) compared with the experimental photos (middle), and overlaid free surfaces (right): (**a**) 0.0*T*; (**b**) *T*/6; (**c**) *T*/3; (**d**) *T*/2.

Figure 4 presents the time histories of calculated pressure at points $M_1$ and $M_2$ by the ISPH method, which are compared with the experimental data. From the contrast, the pressure traces with both first and second peaks calculated by the ISPH method have made good accuracy with experimental results. So the comparisons between the ISPH results and experimental results demonstrate that the ISPH method can get accurate results for the maximum peak values and the phases of the pressure time histories.

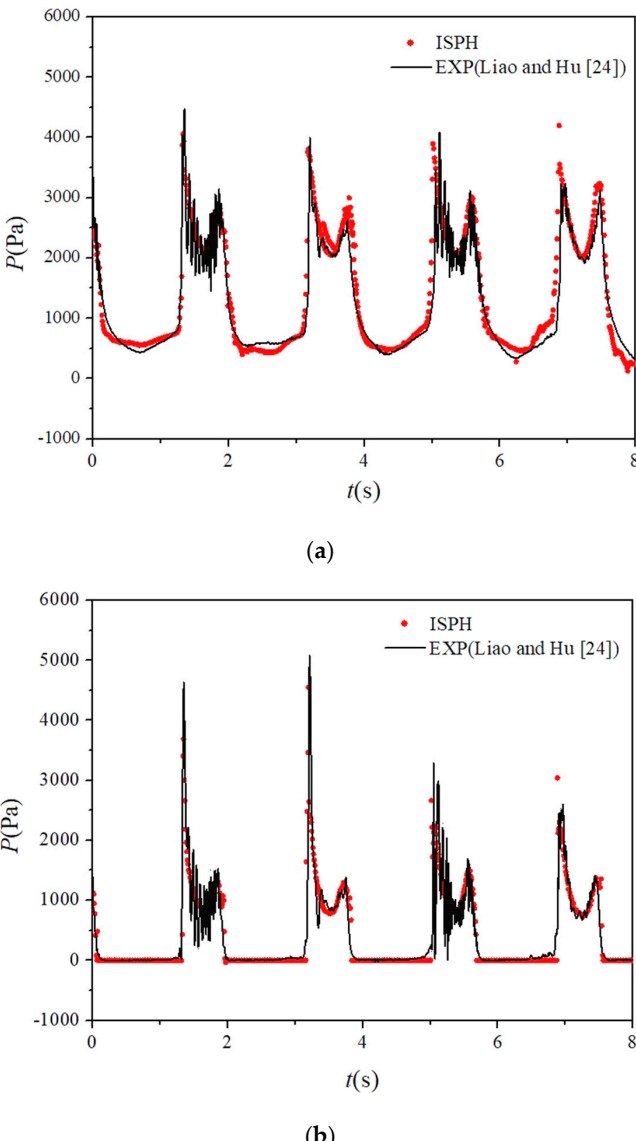

**Figure 4.** Comparisons of pressure time history between ISPH and experimental data of Liao and Hu [24] at: (**a**) $M_1$; (**b**) $M_2$.

### 3.2. Convergence Analysis of ISPH Model

In order to validate the convergence of the ISPH sloshing model, three different particle sizes have been tested, which are 0.002, 0.003, and 0.004 m, corresponding to the number of particles are 49,000, 16,000, and 9000, respectively. And the pressure at $M_1$ and $M_2$ calculated in three different particle sizes are compared with the experimental data, and the results are shown in Figure 5. It shows that the amplitude of non-physical pressure oscillations decreases when particle size is from 0.004 m to 0.002 m. Moreover, the enlarged parts manifest that the pressure traces are more smoothed and more closed to the experimental results. Furthermore, the quantitative comparisons in three different particle sizes are carried on $M_1$ and $M_2$, which obtained the values of the mean error in Equation (13), the results are shown in Table 1, more details can refer to Zheng et al. [23]. From the comparison of pressure results, the errors decrease with the particle number increasing, and the results demonstrate that the ISPH method has very good convergence and stability.

$$E_a = \frac{1}{N} \sum_{i=1}^{N} \left| \frac{\widetilde{P(t)} - P(t)}{P(t)} \right| \tag{13}$$

where *Ea* is the defined mean error; $\widetilde{P(t)}$ is the numerical results of the pressure; $P(t)$ is the experimental pressure; *N* is the number of sampling points.

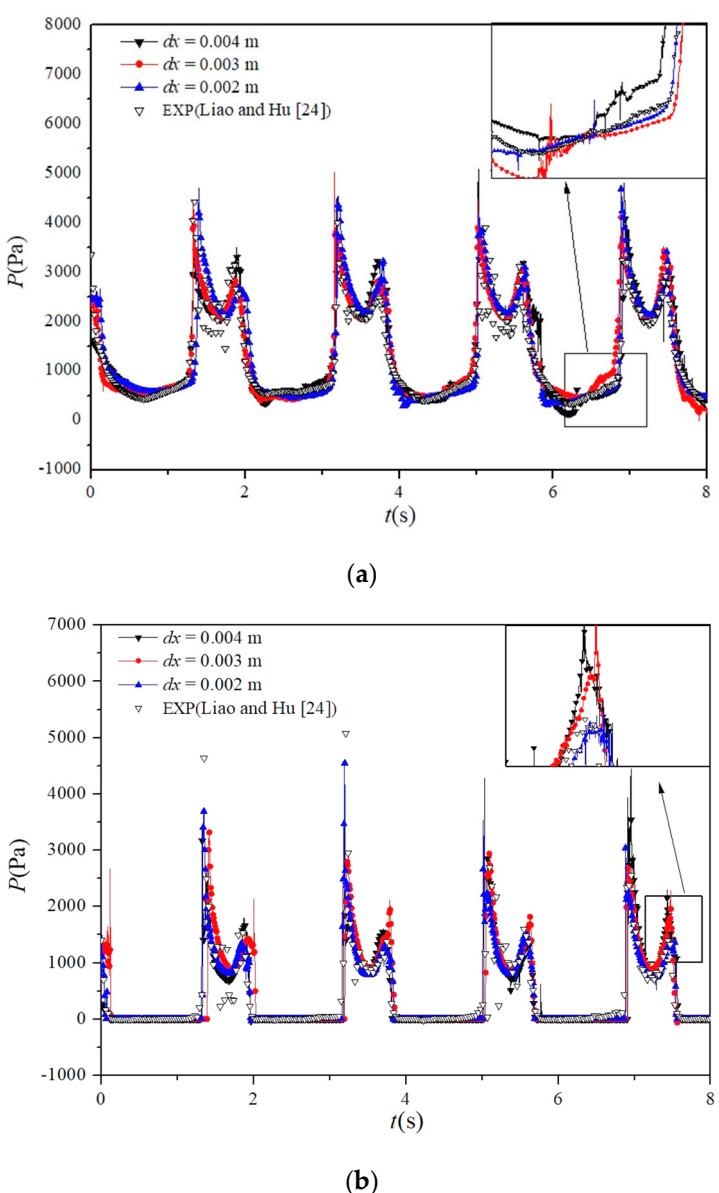

(**a**)

(**b**)

**Figure 5.** Comparisons of pressure time histories between incompressible smoothed particle hydrodynamics (ISPH) and experimental data of Liao and Hu [24] at (**a**) $M_1$ and (**b**) $M_2$.

**Table 1.** Numerical error of ISPH in three particle sizes.

| Particle Size (m) | $E_a$ ($M_1$) | $E_a$ ($M_2$) |
|---|---|---|
| 0.004 | 1.77% | 1.35% |
| 0.003 | 0.97% | 0.93% |
| 0.002 | 0.40% | 0.36% |

Figure 6 gives the convergence rate of the pressure results calculated by this ISPH method. It is shown that the numerical error $E_a$ is closer to the second-order accuracy for the results of pressure calculated at $M_1$ and $M_2$. The ISPH method provides good performance for liquid tank sloshing; thus, it will be used for the study of the sloshing tank applications.

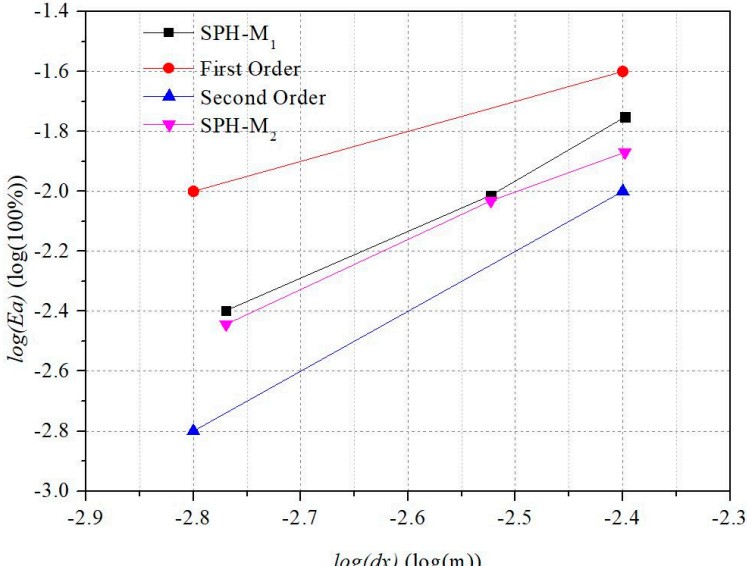

**Figure 6.** Convergence and error analysis of pressure calculated by the ISPH method.

### 3.3. Numerical Validation of the Water Jet Model

Figure 7 shows the schematic view of the water jet test case, which was used in the experiment of Kvicinsky et al. [25]. The diameter of the jet inlet is $D = 0.03$ m, and it is located at $H = 0.1$ m above a flat plate. The initial velocity of the water jet was $v = 19.81$ m/s. In the ISPH, a particle size $dx = 0.0015$ m was used, the time step $dt = 0.00001$ s, and the physical time of simulation was $T = 0.03$ s. The pressures at the plate were recorded.

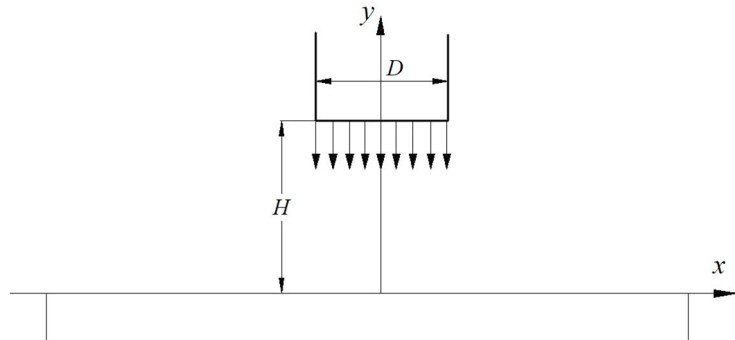

**Figure 7.** Schematic view of the water jet, following Kvicinsky et al. [25].

In the experiment, the pressure coefficient $C_p$ on the flat plate was used as follows:

$$C_p = \frac{2P}{\rho v^2} \tag{14}$$

where $P$ is the pressure at the flat plate, $\rho$ is water density, and $v$ is the initial velocity of the water jet. Figure 8 shows the snapshots of the numerical results. In order to get the stable pressure data, the pressures were extracted when the physical time is greater than 0.015 s. Figure 9 shows the comparison of pressure coefficient $C_p$ among the ISPH results, the experimental data, and the CFD results, where $x$ is the coordinate value on the flat plate. Figure 9 shows that the ISPH results have good agreement with both the experimental data and CFD results, which demonstrates that the ISPH method can have high accuracy on the impact pressure caused by the water jet flow.

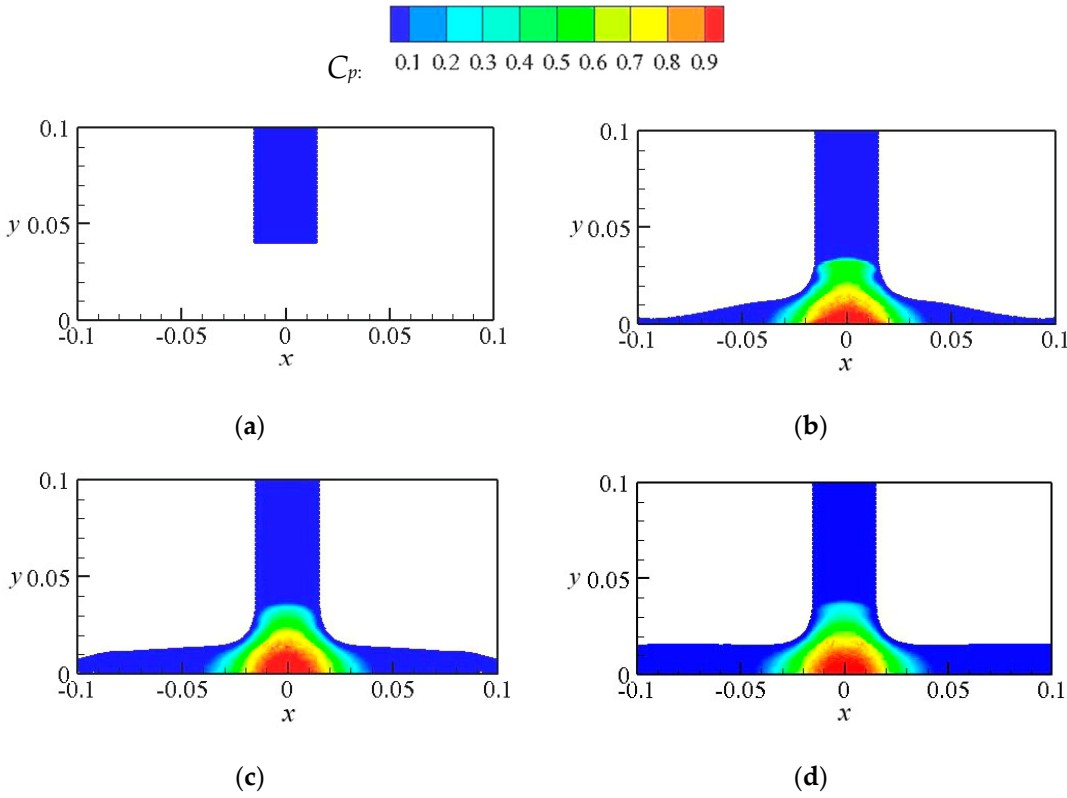

**Figure 8.** Pressure contour snapshots of the water jet case by ISPH: (**a**) *T*/10; (**b**) *T*/4; (**c**) *T*/3; (**d**) *T*.

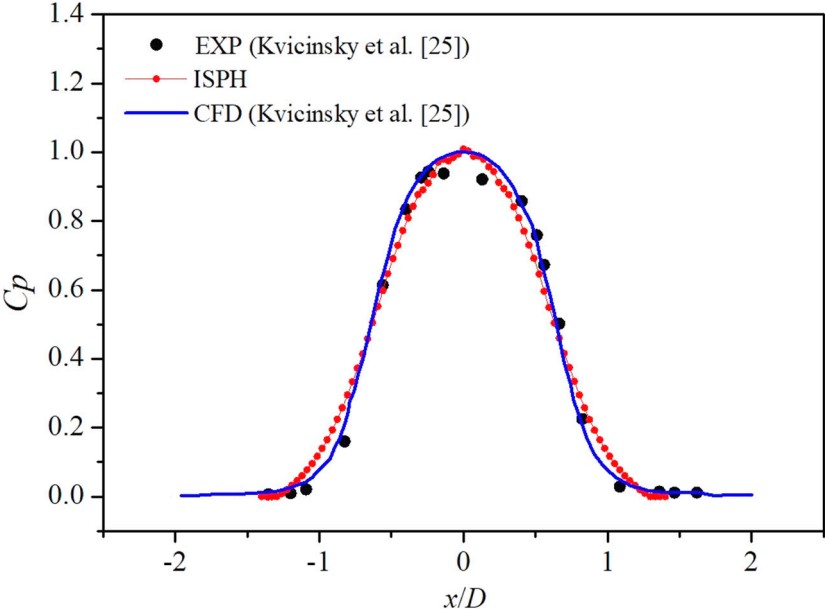

**Figure 9.** Comparisons of pressure coefficient $C_p$ among the ISPH results, the CFD results, and the experimental data of Kvicinsky et al. [25].

### 3.4. Validation of Injected Water Jet Flow Model

Figure 10 gives the geometric dimensions of a rectangular tank. The height of the tank is $H = 0.5$ m, its width is $B = 0.3$ m, and the tank is full of water. $D_i = 0.08$ m, which is the inflow jet diameter in the top center of the tank, and $D_0 = 0.012$ m, which is the outflow diameter. The water jet is injected from $D_i$ with the velocity $V = -0.45\,j$ m/s, and its direction is straight down.

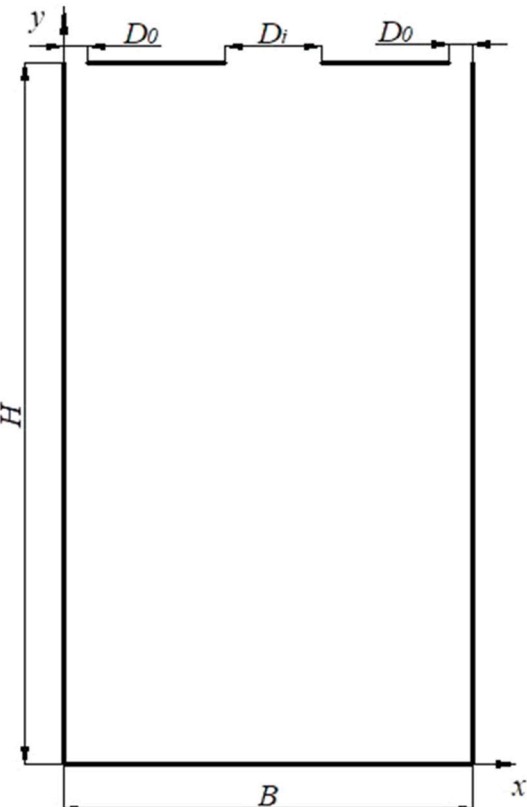

**Figure 10.** Sketch of the jet injected at the top of a water tank.

Figure 11 shows the comparisons of the vertical component of the velocity field $w$ between ISPH and the results of Aristodemo et al. [17] at three significant time instants. It can be seen from Figure 11 that the still water is accelerated progressively with the entry of the water jet flow, and the highest negative velocities are symmetrical by the jet centerline at the initial time. However, these negative velocities become non-symmetrical as time goes on. According to the results of the vertical velocity component by the ISPH method, it is in good agreement with the results of Aristodemo et al. [17].

To quantify the accuracy of the ISPH results, Figure 12 gives the comparisons of relative depth $h/H$ of jet flow between the results of Aristodemo et al. and Fletcher et al. [16,17], where $h$ is the depth of jet flow penetration. The results of ISPH have a high degree of coincidence with the ones of other numerical models. The ISPH method has good potentials for the injected water jet flow.

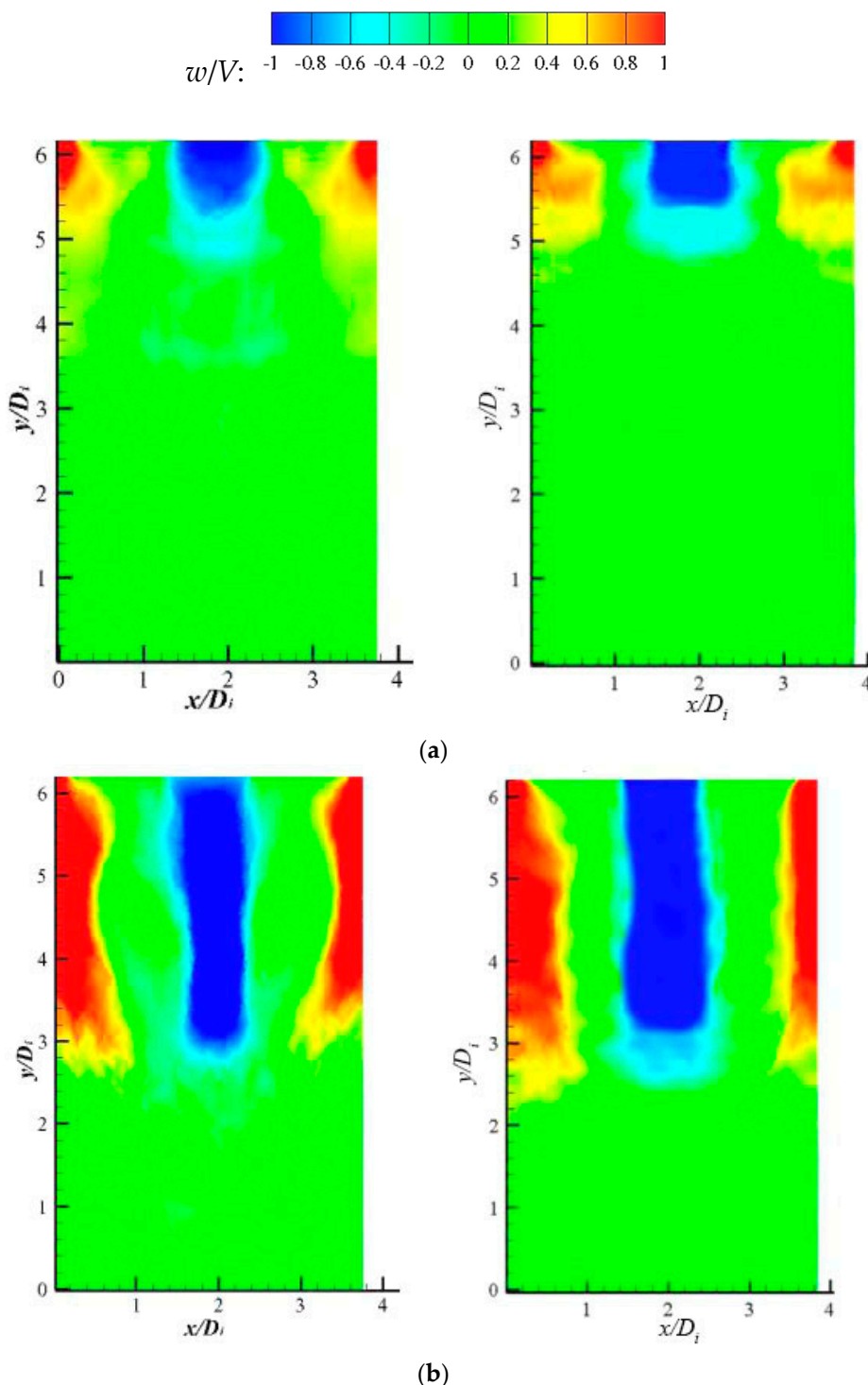

**Figure 11.** *Cont.*

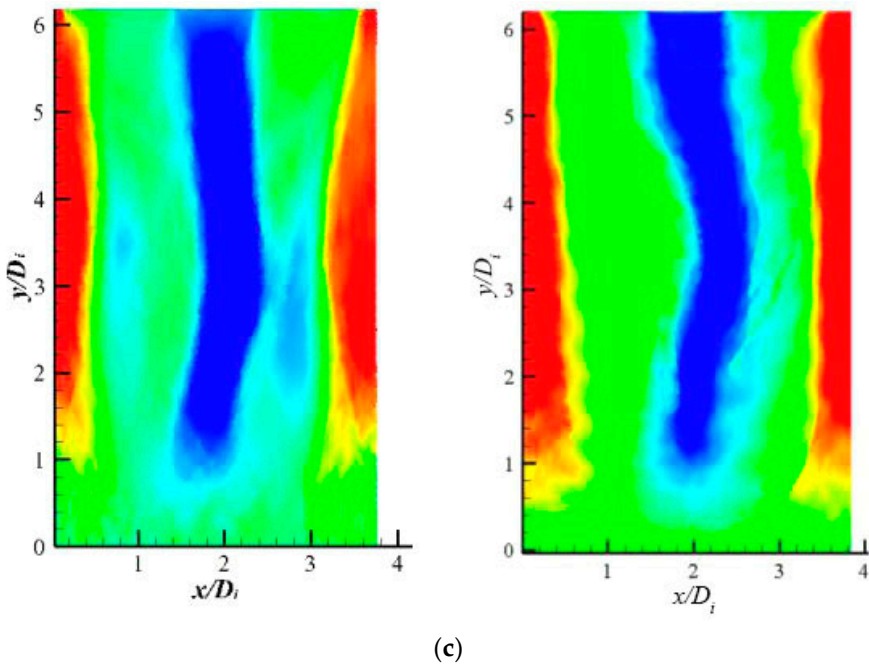

(**c**)

**Figure 11.** Comparisons of the vertical component of the velocity field between the results of the ISPH method (left) and the results of Aristodemo et al. [26] (right) at $t(V/D_i)$ = (**a**) 2.8; (**b**) 6.2; (**c**) 13.4.

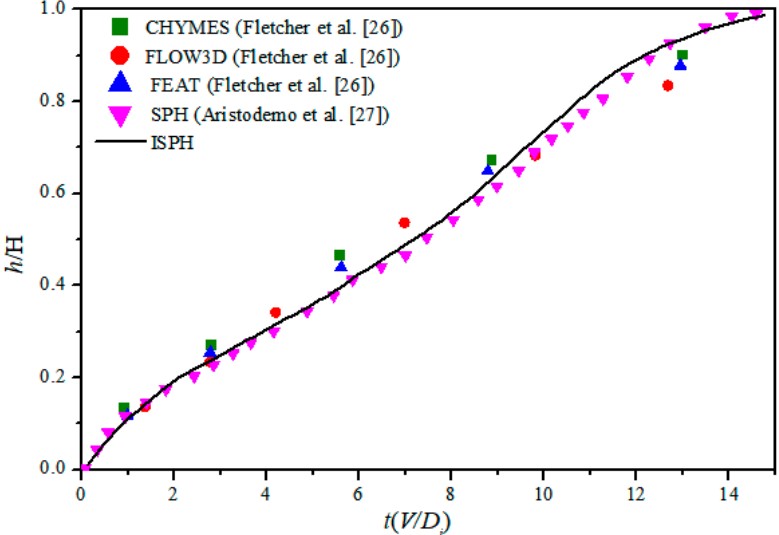

**Figure 12.** Comparison of the relative depth of jet flow penetration between ISPH and other numerical models.

## 4. Results and Analyses

The main scientific problem discussed in this paper is the effect of the water jet on sloshing. The calculation formula of the natural frequency of a liquid tank is shown as follows:

$$\Omega_0 = \sqrt{(g\pi/L)\tanh(\pi d/L)} \tag{15}$$

where $g$ is the gravitational acceleration, $L$ is the length of the tank, and $d$ is the depth of water.

## 4.1. Sloshing Behaviors with Water Jet

In this section, the liquid sloshing in a rectangular tank with a water jet from the top of the tank is considered. The configuration of the sloshing tank is shown in Figure 13, and the geometrical dimensions of the sloshing tank are the same as the tank in Figure 2, but one difference is that there is water flow jetted from the center on the top of the tank. The tank experienced a rolling motion with an amplitude of 8° and an excitation period of 1.85 s. Four sensors, $P_1$–$P_4$, were placed, as shown in the figure, to monitor the pressure distributions. The initial water depth is 0.12 m, and the tank is rotated at the geometric center. The water jet enters the liquid tank through the center on top of the tank, and the initial velocity was imposed to 0.3 m/s.

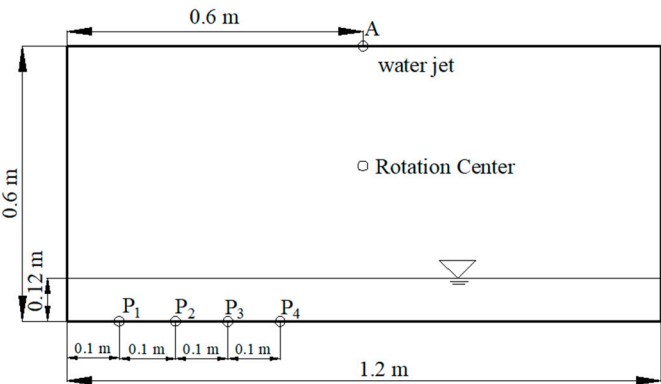

**Figure 13.** Schematic view of the sloshing tank with the water jet from the center on top.

The computed particle snapshots with pressure contours and streamlines for this case are shown in Figure 14. It shows the violent sloshing flow and water jet flow with strong interaction for a sloshing period $T$. The smooth and noise-free stable pressure fields indicate the robustness of the presented ISPH method, which is a good way to analyse the impact pressures. It also can be seen from the streamlines that the fluid is entrapped generating some recirculating counter-rotating cells in the violent sloshing process.

Then, the pressure histories at four different locations at the bottom are recorded and compared. Figure 15 gives the comparison of four pressure histories when the velocity is 0.3 m/s, it shows that the trends of four pressure histories are generally similar, the peak value of impact pressure gradually increases with the water jets into the tank, and the two-peak pressure patterns of four histories are remarkable. By way of contrast, the maximum pressure appears at the location of $P_1$, meanwhile the change range of impact pressure is also the largest. It can get higher pressure when the monitor location is closer to the side wall, the sloshing effect is also more violent in this area. Therefore, in the next cases, the impact pressure at $P_1$ will be studied and analyzed.

The liquid sloshing in a rectangular tank with a water jet of various velocities from the top of the tank is considered. The configuration of the sloshing tank is shown in Figure 13, and different initial velocities of the water jet flow are used in this case, which are 0.2, 0.3, 0.4, and 0.5 m/s, respectively. Figure 16 portrays the snapshots of the sloshing process with pressure fields at $t = 0.296$ s with different initial velocities. It can be seen that when the water jet flow gets into the free surface, the strong impact generates a large free surface deformation with two waves running out along the water jet flow, creating an open-air cavity. Then, the air cavity gradually closes, and the two free surfaces form a short-term bump. From the comparison, the open-air cavity becomes larger with the initial velocity increasing.

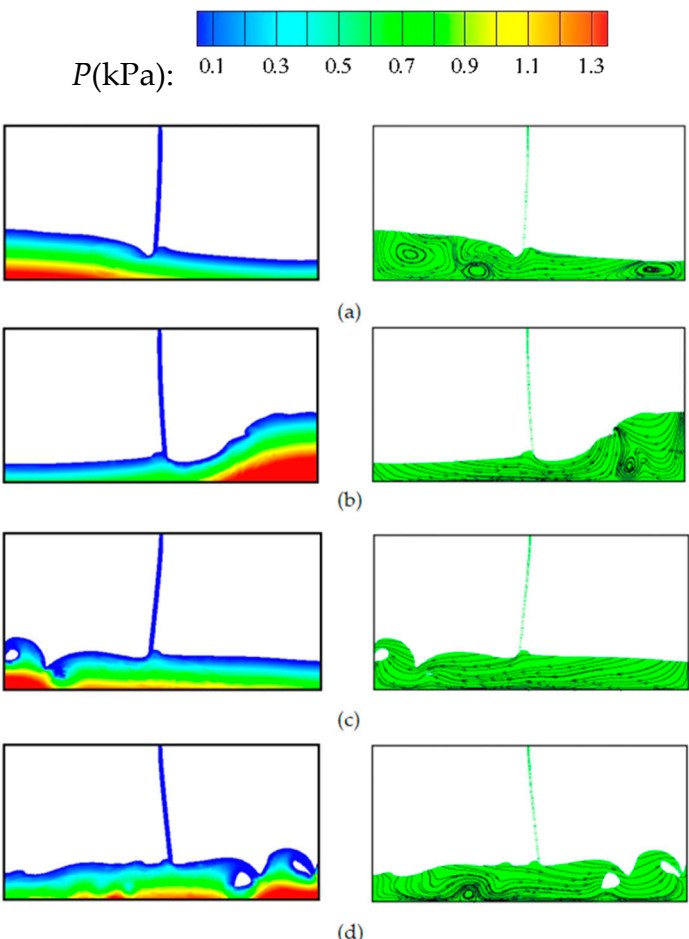

**Figure 14.** Particle snapshots with pressure contours (left) and streamlines (right) of sloshing flow interactions with water jet flow: (**a**) *T*/4; (**b**) *T*/2; (**c**) 3*T*/4; (**d**) *T*.

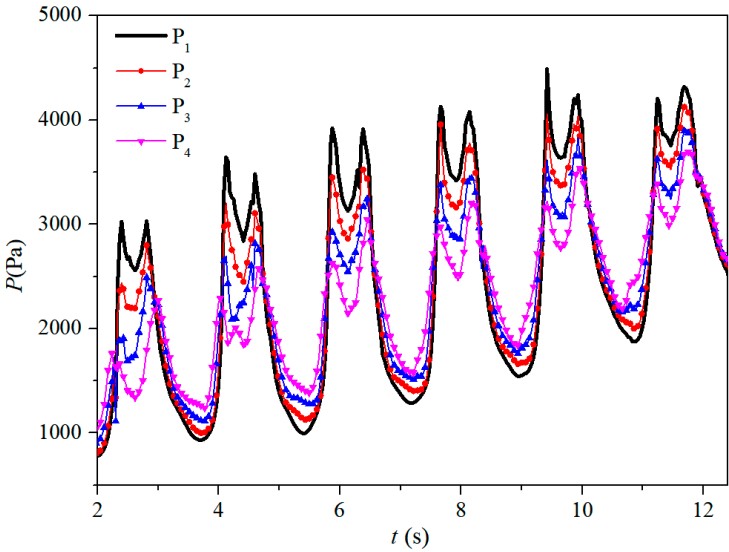

**Figure 15.** Comparison of pressure histories at four different locations at the bottom.

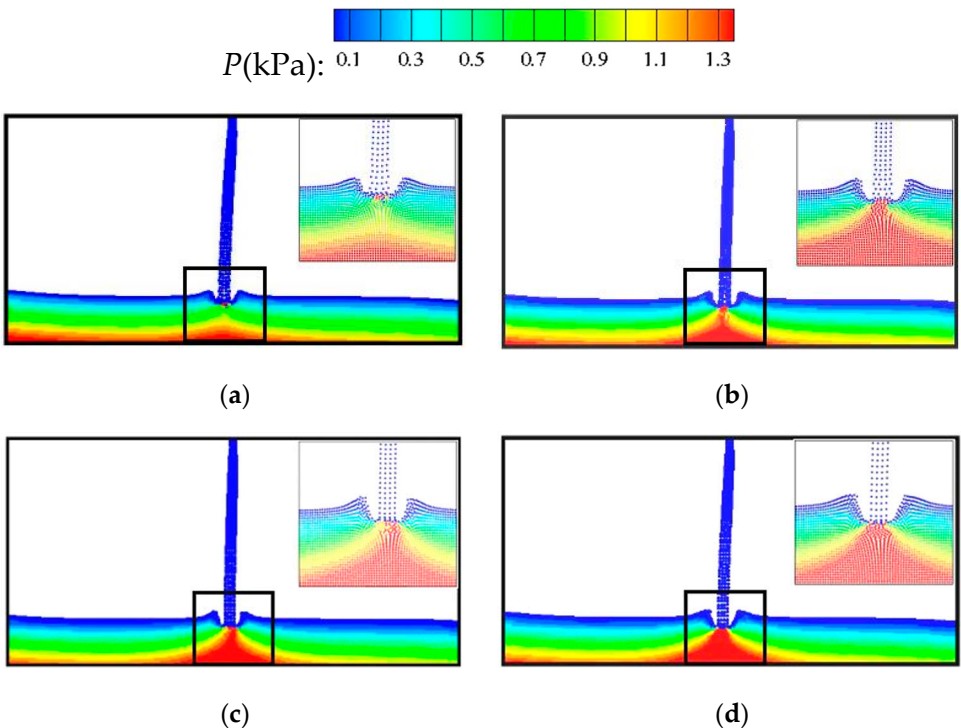

**Figure 16.** Comparison of sloshing patterns with different initial velocities at 0.296 s: (**a**) 0.2 m/s; (**b**) 0.3 m/s; (**c**) 0.4 m/s; and (**d**) 0.5 m

### 4.2. The Effects of the Water Jet Flow Position

Now, the liquid sloshing in a rectangular tank with a water jet flow from various positions at the top of the tank is compared. These correspond to the configurations shown in Figure 17, and there is also a harmonic rolling motion imposed on the geometry center of the tank with an amplitude 8° and a series of excitation frequencies, where $P_1$ is the point of pressure monitoring. $A_1$, $A_2$, and $A_3$ are three different locations for the water flow inlet, and d is 0.5, 0.3, and 0.1 m, respectively, which is the horizontal distance between the pressure probe and the water jet flow. The initial velocity of the water jet flow is 0.3 m/s.

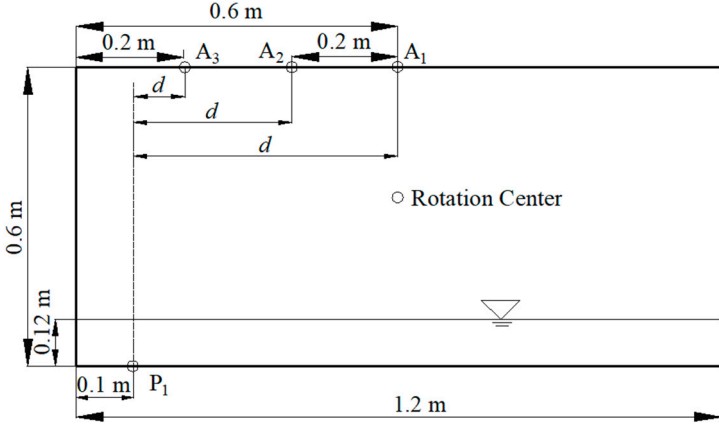

**Figure 17.** Schematic view of the sloshing tank with water jet flows from different positions.

Figure 18 shows the comparisons of free surface profiles at $t = 0.296$ s with the water jet from three positions at the top of the tank when the excitation period is 1.85 s. From the contrast of the three free

surfaces, it can be seen that the deformations of the free surface generated by the impact of three water flows are in a basic agreement, which generates the air cavities with the same shapes and sizes.

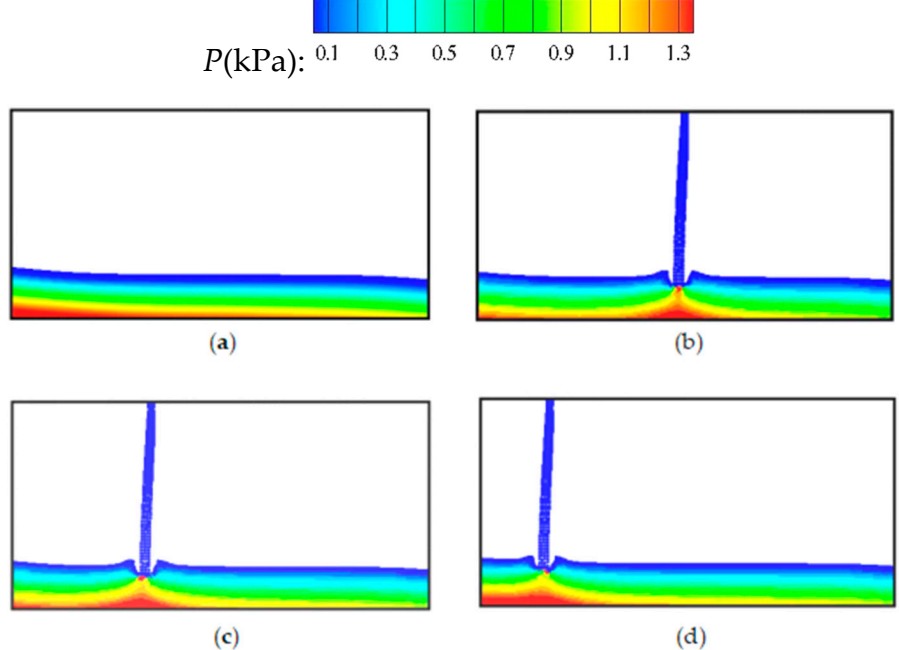

**Figure 18.** Comparison of free surface profiles at the instant $t = 0.296$ s with a water jet from three positions: (**a**) no water jet; (**b**) $A_1$; (**c**) $A_2$; (**d**) $A_3$.

Figure 19 gives the peak value, which is obtained from the impact pressure of $P_1$ at different excitation frequencies. It can be seen that the maximum pressure at $P_1$ decreases when a water flow enters the sloshing tank with an initial velocity, and the excitation frequency where the maximum pressure occurs increases. Also, they change accordingly with the position of the water flow. When the horizontal distance between the water flow and $P_1$ is closer, the effect is more obvious.

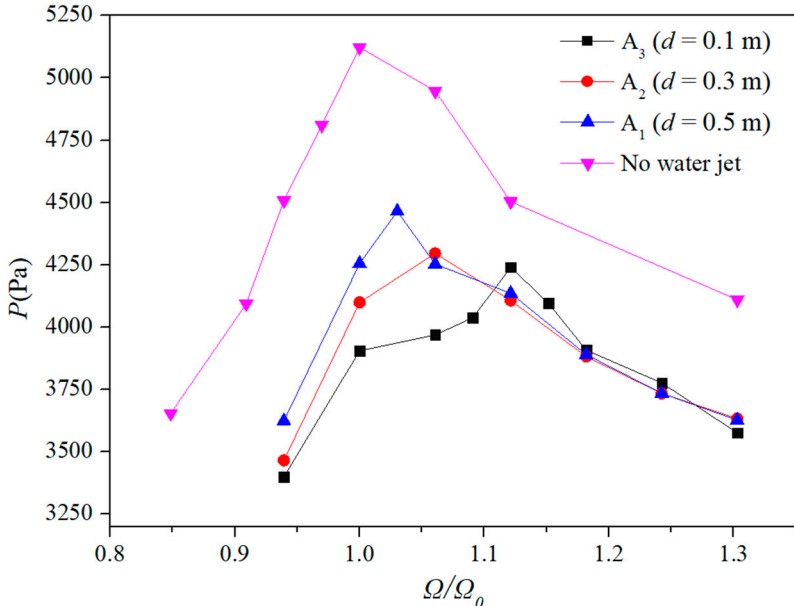

**Figure 19.** Comparisons of impact pressure with the water jet flow in different positions.

Through quantitative analysis, the maximum pressure is reduced by about 12.83% compared to the system with and without a water jet flow at $A_1$, and this difference reaches up to 17.22% when the water jet flow comes from $A_3$. It shows that the water jet flow can reduce the sloshing impact load on the tank wall. When the position of the water jet flow is closer to the pressure probe, the sloshing impact load becomes smaller. In addition, the water jet flow can also change the frequency where the maximum pressure occurs, so it is an effective way of avoiding the appearance of maximum pressure by using a water jet flow in an appropriate position.

### 4.3. The Effects of the Water Jet Flow Number

In this section, the effects of the water jet flow number are investigated based on the configuration shown in Figure 20. The same rectangular tank has a length of $L = 1.2$ m and a height of $h = 0.6$ m, which is partially filled up to an initial depth of 0.12 m. $B_1$, $B_2$, and $B_3$ are three different locations for the water jet flow inlet. Again, a harmonic rolling motion is imposed on the tank with the same amplitude previously used. The sensor $P_1$ is placed, as shown in the figure, to monitor the pressure variations. The same initial velocity of the water jet used in the previous section is also adopted here, which is 0.3 m/s.

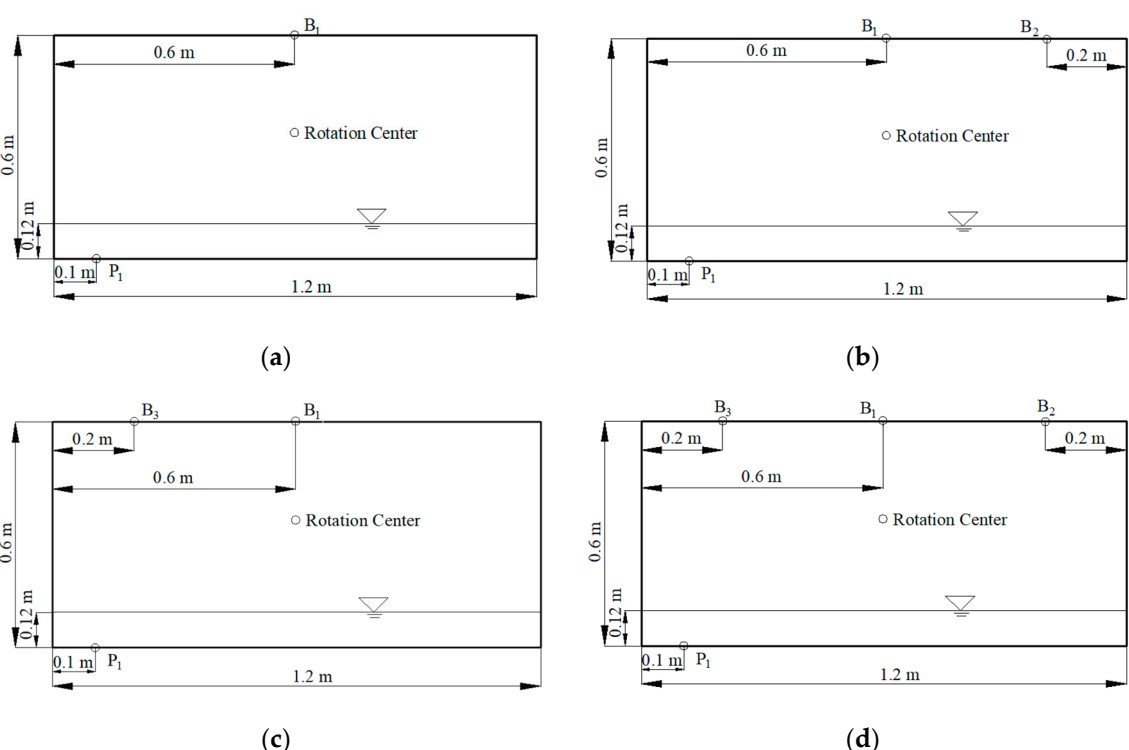

**Figure 20.** Schematic view of the sloshing tank with a water jet: (**a**) only one water flow; (**b**) two water flows at $B_1$ and $B_2$; (**c**) two water flows at $B_1$ and $B_3$; (**d**) three water flows at $B_1$, $B_2$, and $B_3$.

The particle snapshots of the pressure contour for the four cases at $t = 0.296$ s are shown in Figure 21, which illustrates the pattern of the free surface when the water jet flows enter the liquid. It shows that the water jet flow has played an important role in the pressure distribution of the sloshing process because it causes strong collisions between the water jet flow and the free surface below. It should be noted that there are some differences in the pressure distribution at different jet regions.

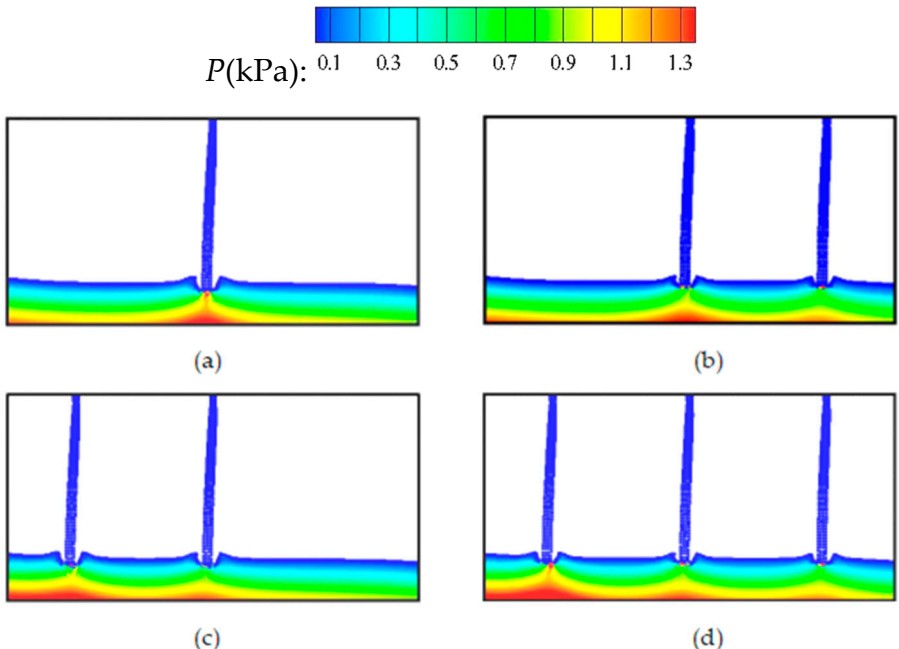

**Figure 21.** Comparison of free surface profiles at $t = 0.296$ s with a different number of water jets: (**a**) only one water flow; (**b**) two water flows at $B_1$ and $B_2$; (**c**) two water flows at $B_1$ and $B_3$; (**d**) three water flows at $B_1$, $B_2$, and $B_3$.

Figure 22 shows the peak value of impact pressure at $P_1$ obtained by ISPH at different excitation frequencies. The comparison includes the case of two water jet flows and only one water jet flow. In all the cases, the excitation frequency is a non-dimensional one, and its value is obtained between 0.9 and 1.5. All the cases correspond to the configurations shown in Figure 20a–c. By comparing the two water flows with only one water flow at $B_1$, it can be seen that the maximum pressure at $P_1$ can get a smaller value when another water flow is added at $B_2$ or $B_3$; meanwhile, the excitation frequency where the maximum pressure occurs also increases. Furthermore, the comparisons between different combinations of two water jet flows are shown in Figure 20b, c. The maximum pressure value of this case combined by $B_1$ and $B_3$ is the smallest. It is reduced by around 5.6% compared with only one water jet flow at $B_3$ and is reduced by around 2.7% compared with the case combining $B_1$ and $B_2$. The excitation frequency where the maximum pressure occurs moves right again when $B_3$ replaces $B_2$. In summary, it demonstrates that the maximum pressure at $P_1$ can be reduced when the other water jet flow is added, and it decreases more remarkably when the water flow is added closer to the pressure probe.

Figure 23 gives the peak value comparisons at $P_1$ for the case of two water jet flows and three water jet flows. The excitation frequency adopts a non-dimensional one, which is between 0.9 and 1.5. The results are obtained according to the configurations shown in Figure 20b–d. The contrast demonstrates that it can obviously be improved for reducing the maximum pressure when more water jet flow is added.

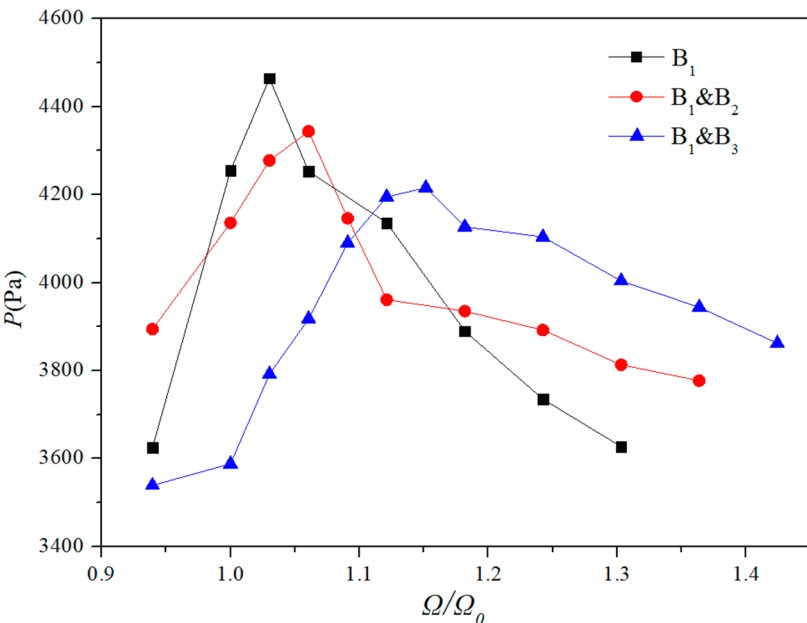

**Figure 22.** Comparisons of impact pressure with the single water jet flow and two water jet flows.

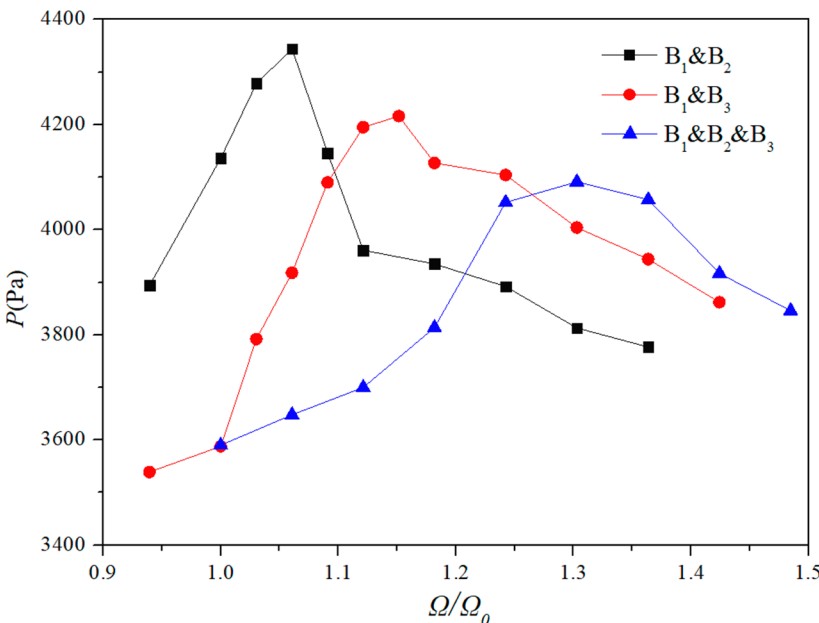

**Figure 23.** Comparisons of impact pressure with two water flows and three water flows.

## 5. Conclusions

In this paper, the incompressible SPH method was used to simulate liquid sloshing in a tank with different configurations of the water jet flow, including the water jet flow with different initial velocities, positions, and water jet flow numbers. ISPH shows good agreement in both the free surface profiles and the impact pressures with the experimental data and demonstrates its great potential in predicting violent sloshing flows. The main purpose of this paper was to study the practical importance of the effect of a water jet flow on liquid sloshing through follow-on model applications of different configurations.

The main conclusions of the paper lie in the following aspects. Firstly, adding a water jet flow at the top of the sloshing tank can reduce the maximum impact pressure effectively. Secondly, adding a

water jet flow at the top of the sloshing tank can change the excitation frequency where the maximum pressure occurs. Thirdly, adding different numbers of water jet flows also can decrease the value of maximum impact pressure. Finally, when the horizontal distance between water flow and pressure probe is closer, the effect on maximum pressure and excitation frequency is more obvious.

Furthermore, it should be noted that there are some limitations in the present sloshing model as follows:

(1) The ISPH computations are based on a 2D model. According to previous experimental research, it seems there is not much difference between the 2D and 3D models, especially in the impact pressure and water surface;

(2) The compressibility of entrapped air also has effects on the violent sloshing process, and the maximum Mach number of all the particles in the violent sloshing process was smaller than 1%, which proves the ISPH model can be used;

(3) In the simulations of coastal and ocean engineering problems, the SPH method is mainly used for the impulsive impact on breaking waves, and the longer simulations are often carried out by traditional CFD methods;

(4) The presence of turbulence would produce fully three-dimensional flow structures in the breaking region at the tip of the wave crest [26]. However, this study focused on the macro liquid impact pressure on the tank walls and the general free surface deformation. Hence, a 2D model could also provide a reasonable simulation.

**Author Contributions:** H.J. and Y.Y. made the computations and data analysis; Z.H. made the data analysis and did the proofreading; X.Z. did the proofreading and editing; Q.M. guided the engineering project and provided the data; Y.Y. drafted the manuscript with others. All authors contributed to the work.

**Funding:** This research work was fundedby the National Natural Science Foundation of China (Nos. 51879051; 51739001; 51579056, and 51639004); Natural Science Foundation of Heilongjiang Province in China (E2018024); Foundational Research Funds for the Central Universities (Nos. HEUCF170104; HEUCDZ1202); Defense Pre Research Funds Program (No. 9140A14020712CB01158).

**Acknowledgments:** The fifth author also thanks the Chang Jiang Visiting Chair Professorship scheme of the Chinese Ministry of Education, hosted by HEU.

**Conflicts of Interest:** The authors declare no conflict of interest.

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
