# Peer review of "Comparative Study on Violent Sloshing with Water Jet Flows by Using the ISPH Method"

_water, doi:10.3390/w11122590_

Round 1

Reviewer 1 Report

The authors studied a two-dimensional incompressible smoothed particle dynamics method to simulate sloshing in a tank with various jet flow parameters such as impact pressure, number of jet flow, etc. They showed a good agreement between ISPD and experimental measurements.

The del operator is a vector operator, and it should be mentioned in the text, or an arrow should be placed on top.

A nomenclature part is missing.

What is mj in equation 3? What is h in equation 4? How does (del)i is defined in equation 5? What is u star in eq. 6? What does ()0 and ()* represent? What is the value of eta (eq. 11) in the simulations?

The pressure is shown using the lowercase p. Later in section 3.1, the authors used P1 and P2 as pressure sensors. It will create confusion for the reader.

Please explain how do you choose the values for dx and dt in section 3.1. What is the effect of these parameters on the solution?

E_a in eq 13 should be written as Ea

In equation 14, what does the term v represent? Vertical velocity!? should be explained in the text and the nomenclature.

Line 242: velocity is a vector and should be adequately defined, such as V = - 0.45 j

The legend of Figure 11 is misplaced.

The legend of Figure 14 is misplaced.

To be consistent, the title of Section 3 should be fixed. "results"->"Results."

I recommend the manuscript for publication if the authors fix the above minor issues.

Regards,

Author Response

Thank you very much for constrcutive suggestions, the details of the point-by-point response are in the attachment, please kindly check the attachment.

Reviewer 2 Report

This paper adopts the ISPH to study the violent sloshing with water jets. Some numerical techniques are considered, like a water jet inlets, free surface detection method and poison pressure equation solver. The manuscript is well written and seems to be acceptable for the journal of Water, which is mainly for the case studies adopted to study the performance. However, there are some difficulties in appreciating the results from the comparisons to show the better performance of some improved numerical technologies. Some of them can be found in many places in the comments.

In the part of Inlet boundary treatment, it is not clear to explain the process of inlet particle handling, it should give the explanation of Fig.1 at different steps.

As for the scientific work, do they (i) want to promote their method or (ii) want to study an effect of practical importance? It sounds as if they want to do both simultaneously, please give the explanation.

The authors state in line 43-46 should be explained in more detail. What is the difference in handling the free surface? And how they are implemented in different ways with respect to different methods.

What is Alpha in Equation (8) tuned to, and how is it tuned? Please elaborate.

"divergence-free" and "density-invariance" in line 110 are not read as clearly related to the two-step approach described previously: please make it so if that is the case.

The comparison of the surface in Figure 3 is only qualitative and does not fully support the good agreement of the two methods. I suggest the author to overlay the surface obtained by the two methods.

There are small typos and mistakes in the paper:

On page 1, it shouldn’t be “Water 2017, 9, x; doi: FOR PEER REVIEW “ on line 237, there is a blank line. Some pictures such as Figure 14. should be adjusted on one page.

Author Response

Many thanks for referee's many good suggestions and comments, the details of the point-by-point response to reviewer's comments are in the attachment, please kindly check it.

Round 2

Reviewer 2 Report

I am happy to see that all my comments have been address properly and would like to recommend for publication in the present form.